# Health Impact Assessment (HIA) of a Daily Physical Activity Unit in Schools: Focus on Children and Adolescents in Austria Up to the 8th Grade

**DOI:** 10.3390/ijerph19116428

**Published:** 2022-05-25

**Authors:** Madlene Movia, Sandra Macher, Gabriele Antony, Verena Zeuschner, Gabriela Wamprechtsamer, Judith delle Grazie, Helmut Simi, Bianca Fuchs-Neuhold

**Affiliations:** 1Institute of Health and Tourism Management, FH JOANNEUM—University of Applied Sciences, 8344 Bad Gleichenberg, Austria; madlene.movia@gmail.com (M.M.); sandra.macher2@fh-joanneum.at (S.M.); helmut.simi@fh-joanneum.at (H.S.); 2Austrian National Public Health Institute, 1010 Vienna, Austria; gabriele.antony@goeg.at; 3Healthy Austria Fund, 1020 Vienna, Austria; verena.zeuschner@goeg.at; 4Federal Ministry Republic of Austria—Social Affairs, Health, Care and Consumer Protection, 1010 Vienna, Austria; gabriela.wamprechtsamer@bmg.gv.at (G.W.); judith.dellegrazie@gesundheitsministerium.gv.at (J.d.G.)

**Keywords:** health impact assessment, physical activity, children, adolescents, pupils, school setting, Austria

## Abstract

Physical activity has a variety of health benefits for young people. The World Health Organization (WHO) recommends that children and adolescents aged 5–17 years should be physically active for at least 60 min a day. This Health Impact Assessment (HIA) examined the potential impact of a daily physical activity unit in Austrian schools, with a focus on children and adolescents up to eighth grade. The HIA methodology systematically followed several stages: screening and scoping, appraisal, and formulation of recommendations. A literature review, an online survey (*n* = 712), focus group discussions (*n* = 4), and appraisal workshops (*n* = 2) have been carried out. The analyzed results indicate a primarily positive impact on the individual health of children and adolescents, on social and community-related networks, on living and working conditions, and on the system level. Recommendations for the implementation include ensuring equal opportunities and support for disadvantaged children and schools. Furthermore, the comprehensible selection of pilot regions and the nationwide resource planning should be considered. Additional important factors include the education and further training of pedagogical staff and coaches, and the availability of sufficient infrastructure. In the long term, the fixed integration of five additional teaching units of physical activity per week, and an increased focus on the elementary/nursery school sector were identified as significant.

## 1. Introduction

According to the World Health Organization (WHO), physical activity has a wide range of health benefits for hearts, bodies, and cognitive abilities. It enhances the prevention of noncommunicable diseases, e.g., cancer, diabetes, or cardiovascular diseases, and helps to maintain a healthy body weight. Physical activity has further positive effects on mental health, e.g., depression or anxiety. The importance of physical activity is that it ensures healthy growth and development in children and adolescents [1]. However, physical inactivity is a public health hazard worldwide. A study about global trends in physical activity showed that 81% of children and adolescents worldwide (aged 11–17 years) are insufficiently physically active [2]. The WHO guidelines [3] and the Austrian physical activity recommendations [4] indicate that children and adolescents (aged 5–17 years) should perform at least 60 min moderate-to-vigorous intensity physical activity per day. In Austria, 83% of girls and 73% of boys (aged 11 to 13 years) do not achieve the recommendations of at least 60 min of physical activity per day [5]. In addition, 13% of Austrian girls and 20% of Austrian boys (aged 10 to 14 years) are either overweight or obese [6].

Already in 2018, the Austrian Parliamentary Directorate issued a request for the promotion of physical activity initiatives in schools and the expansion of daily physical activity units for all children and young people of compulsory school age [7]. Currently, the daily physical activity unit is not yet implemented at compulsory schools in Austria. However, in the current Austrian government program (2020–2024), the “expansion of physical activity and sports within the school curriculum” was stated and it was specifically mentioned that a daily physical activity unit should be included in the Austrian school setting. In order to address the problem of physical inactivity in Austrian schools, the Federal Ministry of Education, Science and Research and the Federal Ministry of Arts, Culture, Public Service and Sports commissioned an interdisciplinarity project working group to develop a concept for the “Daily Physical Activity Unit” in schools.

On behalf of the Federal Ministry of Social Affairs, Health, Care and Consumer Protection, FH JOANNEUM—University of Applied Sciences, Institute of Health and Tourism Management, conducted a Health Impact Assessment (HIA) on the introduction of the daily physical activity unit for school children in Austria from the first to the eighth grade. To the authors’ knowledge, the present HIA is the first of its kind performed in Austria and therefore provides new insights to address the problem of physical inactivity among children and adolescents in Austria.

## 2. Materials and Methods

The process of the present HIA was devised based on the main stages of the HIA methodology, which include, according to the WHO screening and scoping, appraisal, and formulation of recommendations [8]. The present HIA does not include the “monitoring and evaluation” phase as results should be available promptly. Furthermore, various methods have been used to gather the information: a systematic literature review and analysis, an online survey, focus group discussions, and appraisal workshops (see Table 1).

### 2.1. Screening and Scoping

The goal of the screening phase was to define the scope and boundaries of the present HIA. An advisory panel was established to discuss basic framework conditions such as prioritization of topic areas, focus (concept, setting, target groups, methodology, timeline), the definition of a uniform type of communication, and distribution of roles during a planning workshop.

### 2.2. Appraisal

The appraisal phase of the present HIA was built on the determinants of health [9], that can influence the health of children and adolescents, and have been appropriately tailored to the purpose of this HIA. The determinants of health of the present HIA include the following parts: individual health; social and community-related networks, impact on further target groups; living and working conditions (school-setting); and general conditions of system, socio-economic, and environmental factors.

For this purpose, a selective literature review was conducted in relevant databases (PubMed, Cochrane, SPOLIT, Science Direct). In addition, the following data collection methods were used to identify and analyze potential health effects:

**The literature search** and further processing of studies, case studies, best practice examples, and data was primarily carried out by a selective literature search in relevant databases (PubMed, Cochrane, SPOLIT, Science Direct). The main search terms considered and combined in the literature review included: physical activity, physical education, daily exercise, health promotion, school, education, child, adolescent, migrant background, equal opportunities, disadvantaged, disabled child, girls, health outcome, health impact (assessment). Additional literature was identified through a snowballing approach in which further references of studies were identified through the reference list. The selection criteria for the literature search were also based on the determinants of health of the present HIA. Therefore, articles had to include findings on physical activity in relation to the individual health of children and adolescents; social and community-related networks, impact on further target groups; living and working conditions (school-setting); and general conditions of system, socio-economic, and environmental factors. The studies were selected based on a high-quality standard. Primarily, study types such as systematic reviews, meta-analysis, and randomized controlled trials (RCTs) were selected. Longitudinal and cross-sectional studies as well as national programs, reports, and articles were also included in the literature synthesis if the content, methodology, and conduct were sound. The literature search was conducted between April and May 2021. The search was limited to English and German language publications. Suitable literature was entered into an Excel spreadsheet, screened, and summarized. Sources were grouped by author, title, study type, content, and annotations. After assigning them to one of the four health determinants, the HIA project team individually assessed the relevance of the sources found. All literature sources were then discussed, and a collective decision was made to include or exclude the source.

**The online survey** instrument was initially created by the HIA project team. Subsequently, verbal and written feedback was obtained from the extended HIA team and the HIA advisory board. Additionally, two pre-tests were conducted. Feedback was incorporated on an ongoing basis. The final survey included a total of 38 questions that were entered into the online survey tool “Lime Survey”. The survey link was accessible from May 28, 2021, to June 21, 2021. The survey link was distributed through the HIA advisory board, two region-wide newsletters, and social media channels. The HIA project team analyzed the collected data using SPSS—Statistical Package for the Social Sciences [10] and Microsoft Excel [11] programs. A total of 712 fully completed questionnaires were included in the analysis.

Based on the results of the online survey, four target group-specific **online focus group discussions** were conducted with different participants to obtain more in-depth information on the impact of the daily physical activity unit (see Table 2). 

The discussions took place between 15 June and 22 June 2021 and were conducted via the online platform Microsoft Teams [12]. The potential impact on the target groups (children, teachers, parents, etc.) was discussed and recommendations for the daily physical activity unit were collected. The conversations were recorded and afterward summarized and sent to the participants. Subsequently, the discussions were anonymized, analyzed, and evaluated. To capture children’s opinions, parents or legal guardians were given guiding questions to ask their children in advance and then share the information gathered in the focus group.

For the **appraisal workshops**, all results of the literature search, the focus group discussions, and the online survey were summarized and presented in an assessment table. Based on the four predefined determinants of health, each assessment sheet contained detailed information on identified health impacts. A consensual decision with all participants was made on a five-point Likert scale regarding positive or negative effects and the probability of these impacts occurring (low-medium-high) (see Figure 1). Workshop participants included members of the advisory board, parent representatives, experts from the fields of health, education, and sports, and other relevant stakeholders. The first three-hour online workshop took place on 29 June 2021, and another two-hour online workshop was held on 7 July 2021.

To ensure **informed consent and data protection**, an information sheet and a consent form were sent to all participants prior to the focus groups and appraisal workshops. All recorded files were deleted after anonymized transcription.

### 2.3. Formulation of Recommendations

The fields of action and associated recommendations were prepared in a first step by the HIA project team. Feedback loops were conducted with the advisory board and revised accordingly. The final report was submitted to the Federal Ministry of Social Affairs, Health, Care and Consumer Protection at the end of September 2021.

## 3. Results

The results are categorized and presented according to the health determinants of the present HIA, which include again the following parts:individual health.social and community-related networks, impact further target groups.living and working conditions (school-setting).general conditions of system, socio-economic, and environmental factors.

### 3.1. Individual Health

Several studies have shown that daily physical activity, throughout childhood has several positive effects on health: physical, psychological, and social [13]. In addition to sport motor skills [14,15], these include cognitive development and problem-solving skills [16,17], academic performance [18,19], and psychological well-being [20,21]. In addition, recent research indicates a positive relationship between physical activity in school and physical activity in leisure time [22].

The participants in the focus group discussions and the online survey also agreed with these findings. In addition, the majority of respondents indicated that the daily physical activity units can have (very) positive effects on stress management (stress management) and violent/aggressive behavior.

Overall, the impact of the program on the individual health of school-aged children was rated as positive.

#### Recommendations Referring to Individual Health

**Joy of movement as the top priority:** The joy of regular movement should be the main focus. Needs-based concepts should be developed and offered that include an optimal selection of movement games and non-competitive activities that positively influence children’s psychological well-being, self-confidence, social cohesion, and mutual appreciation.**Individual health and inclusion:** To promote individual health, attention should be paid to motivational factors (e.g., “nudging”), the absence of team selection procedures, fair play, and culturally and gender-sensitive conditions. In addition, further reflection and conceptualization are needed on how to motivate and adequately involve groups that require special attention (children who do not like exercise, children with physical and cognitive impairments, etc.).

“*Exercise is beneficial for school success and learning*.” (Quote, parent focus group)

### 3.2. Social and Community-Related Networks

Daily physical activity can be beneficial to social health in childhood [23]. School-based interventions that involve families and the community can also have a positive impact on the physical health of children and adolescents and significantly influence physical activity levels [24]. This is also confirmed by the present survey results, which add that factors such as family climate, friendships, development of social behavior, reduction of social isolation (loneliness), quality of life at school (school climate), relationships with classmates and teachers can be (very) positively influenced by such an intervention.

Furthermore, studies confirm that girls exercise significantly less than boys of the same age [25]. In addition, children from migrant families spend less time being physically active than children without a migrant background [26]. In addition, children with cognitive impairments (including sports activities) participate less in social life than their peers [27]. According to the online survey, respondents consider children who are overweight or obese (66%), children with a migrant background (39%), girls (37%), boys (33%), children from low-income families (32%), children with limited mobility (27%), children with cognitive impairments (24%) as target groups that should be given special attention when implementing the daily physical activity unit.

Overall, the impact of the program on social and community-related networks of children of compulsory school age was assessed as positive. Factors to be taken into account in this area include equal opportunities and support for disadvantaged pupils and schools.

#### Recommendations Referring to Social and Community Related Networks

**Social behavior and participation:** The planned program have great potential to positively influence the development of children’s social behavior, and the social environment as well (family, friends). In this way, the project can specifically prevent discrimination tendencies in the field of physical activity and sports at schools and promote inclusion. The idea of cooperation with the combination of fun, joy, voluntariness, co-determination, and support can serve as a motivational impulse for children.**Particularly affected groups:** There is a need for sensitive handling of indicators used to assess the need for physical activity in children (e.g., body mass index was seen critically as an indicator). Furthermore, adequate communication strategies (e.g., a support system for affected children should be offered and communicated to target groups) and training concepts (e.g., peer approaches, focus on gender sensitivity, special training for dealing with particularly affected children) are necessary to motivate all children to be more active and to counteract stigmatization. For children with body image deficits, strategies and alternatives could be developed to protect them from shaming and bullying, e.g., by providing physical activity programs for overweight children. To create equal opportunities for socioeconomically disadvantaged children and their families, the program should be offered free of charge to all children. In addition, the involvement of stakeholders from neighborhoods and communities and the use of community networks (e.g., clubs) is considered an important factor in ensuring that children and adolescents are exposed to a variety of different types of exercise and sports. Moreover, care is thus taken to ensure that sufficient infrastructure can be provided for the practice of sports (including in public spaces).

“*Exclusion can only be avoided by more movement for everyone in the class, so children with mental or physical disabilities should also be encouraged to exercise more*” (Quote, focus group with specialist inspectors for physical activity and sport in schools)

### 3.3. Living and Working Conditions

Studies confirm that schools can contribute to 40% of children’s physical activity. Physical activity intervention programs and more sports as well as adequate equipment in schools lead to an increase in physical activity among school-aged children and adolescents [28]. In addition, the work or school environment has an impact on sedentary behavior. For example, standing desks in the work environment and encouraging stair climbing is also an important factor in reducing sedentary behavior [29].

The pedagogical quality in the areas of physical activity, school life/social learning (school climate), job satisfaction, and education and training of school staff can be positively influenced by the implementation of a daily physical activity unit. Regarding the additional personal effort of the pedagogical staff, 45% of the participants of the online survey indicated neutral or no effects, 30% (very) positive effects and 20% suspected (very) negative effects. In addition, the majority indicated a (very) positive impact on break rooms/yards, open spaces/playgrounds, and exercise-friendly school room designs. Parents/guardians or parent representatives agreed that the daily physical activity unit can have a (very) positive impact on family satisfaction, individual recreation time, and family leisure time behavior.

In addition, throughout the online survey, respondents were asked about ways to implement the daily physical activity unit. The results show that respondents were in favor of the following factors:The integration of the daily physical activity unit (five lessons per week) as a fixed component in the curriculum (77%),the implementation of sport-specific offers in cooperation with sports clubs (71%),integrating physical activity into everyday school life outside of lessons on physical activity and sports (64%), andthe promotion of active mobility on the way to and from school (55%) andthe implementation or addition of a daily physical activity unit by external physical activity coaches during class time (49%).

Almost half of the respondents stated that all-day school offerings do not conflict or compete with leisure clubs (especially sports clubs). In addition, 62% are in favor of integrating sports clubs more strongly into everyday school life in the future.

Overall, the impact of the program on the living and working conditions of children of compulsory school age was assessed as positive. Factors to be considered in this area are commitment, organization, and movement infrastructure.

#### Recommendations Referring to Living and Working Conditions

**Education and Training:** Educational quality in the area of physical activity can be improved through the daily physical activity unit. Examples include academic education and training of teachers in physical activity and sports, strategies for bullying prevention, and implementation of incentive systems and methods to promote physical activity. The training of elementary school teachers should be expanded with regard to physical activity and sports content. Pedagogical knowledge and didactics are important for the training of physical activity instructors in order to respond to the different needs of children. Mutual learning between educators and movement trainers can also be considered (e.g., to increase the motivation and initiative of both groups). In addition, physical education students can also be involved in the process of lesson design.**Structural, financial, and personnel conditions:** Due to a possible increase in school-internal care time it is necessary to deal with structural, personnel, and individual framework conditions. An increase in the number of teachers, sports infrastructure such as gymnasiums, sports equipment, scheduling of breaks, transportation (bus schedules, pick-up, and drop-off services), etc. should be considered and coordinated. Furthermore, different conditions in urban and rural areas are to be expected. This regional diversity can be a barrier or extra work for schools and educators and should be taken into account to avoid negative consequences in terms of acceptance and feasibility.**Mandatory vs. Voluntary:** In terms of voluntary or mandatory implementation, one solution strategy could be to design and implement a hybrid system consisting of both voluntary and mandatory parts. A voluntary implementation of the project might not achieve the optimal effect for children, who would benefit greater from more physical activity. In addition, pupils’ participation in physical activity and sports in the school environment should be encouraged. In addition, the services should be free of charge in order not to burden the family budget.**School physician system:** The school doctor system could also contribute to the project (e.g., detection of postural defects, etc.).**Early promotion:** In order to promote exercise in children as early as possible, it is advisable to anchor measures and concepts already in the elementary education sector (e.g., nursery school and children’s groups).

“*Most of the spatial infrastructure would have to be improved. Several schools often share only one gym*.” (Quote, focus group school management and teachers)

### 3.4. System and Environmental Factors

According to Ding et al. (2016), physical inactivity is responsible for a substantial economic burden worldwide [30]. An international analysis by Hafner et al. (2019) indicated that making people physically more active is associated with economic benefits [31].

More than half of the participants in the online survey also agreed that (very) positive effects on relieving the burden on the health care system, on equal opportunities, and on economic activity (more consumption, e.g., purchase of sportswear, training costs for educators) can be expected. The majority of respondents also expect (very) positive effects on the use of green spaces and water areas or the use of other recreational facilities and infrastructures (including parks and playgrounds), the use and safety of bike paths and sidewalks, and consequently an improvement in air quality.

Overall, the impact of the program on the system and environmental factors of children of compulsory school age was assessed as positive.

Factors to be considered in this area are the selection of pilot regions for the implementation as well as the future nationwide financing.

#### Recommendations Referring to System and Environmental Factors

**Active mobility:** To promote active mobility, it is important to increase safety on routes to school. Stakeholders who have an influence on transport infrastructure, as well as parents, should be made aware of and involved in activities to promote active mobility in childhood. Examples mentioned were “pedibus”, bicycle cooperatives, walking with neighbors or grandparents/elderly people, and docking with health-promoting projects in communities/districts.**Infrastructure:** infrastructural framework conditions, such as the creation of physical activity spaces, should be discussed in cooperation with the municipalities/counties. Promotion of outdoor exercise should also be considered.**Pilot regions:** It is recommended to systematically select pilot regions according to predefined criteria to obtain a representative picture of the Austrian compulsory school landscape (urban/rural, school type and size, location and infrastructure, groups to be considered). Target indicators can be defined for the subsequent standardized evaluation.**Provision of resources:** In order to be able to implement the project uniformly throughout the nation, resources should be provided at the personnel and financial levels. Established country-specific measures should be considered in the implementation and thus synergies should be used in a targeted manner.

*“Sports-based school projects often fail because of the “cost” justification, and for that reason, I am pleased to be here today participating in this focus group.”* (Quote focus group school leadership)

## 4. Limitations

First, it should be mentioned that children’s and adolescents’ views were gathered only indirectly. Therefore, parents or legal guardians were asked to interview their children based on predefined questions and then share the information gathered in the focus group discussions. Further research should include a direct collection of information from children and adolescents. Moreover, the present HIA was subject to time pressure, as the results and recommendations had to be available promptly. Another limitation refers to the focus group discussions and appraisal workshops, which took place online rather than in person.

## 5. Conclusions

To the authors’ knowledge, the present HIA is the first of its kind performed in Austria. The target of this HIA was to assess the potential impact of a planned daily physical activity unit in Austrian schools, with a focus on children and adolescents up to the eighth grade. Based on a literature review, an online survey, focus group discussions and appraisal workshops, primarily positive effects on the individual health of children and adolescents, on social and community-related networks, on living and working conditions, and on the system level, could be identified.

Factors to be considered when it comes to the implementation of the physical activity unit, firstly, refers to equal opportunities and to the support of (potentially) disadvantaged children and schools. Secondly, aspects of the binding nature/voluntariness of the physical activity units should be taken into account. In addition, high-quality implementation of the physical activity units should be ensured. This includes the training and education of teachers and coaches, the consideration of organizational and structural differences of the respective school types and regions, as well as the availability of sufficient infrastructure. Furthermore, the comprehensible selection of pilot regions and the nationwide resource planning, represent serious factors. In the long term, the fixed integration of five additional teaching units of physical activity per week, as well as an increased focus on the elementary/ nursery school sector, was identified as significant.

The present HIA is based on majority decisions, assuming that the planned project is implemented in a quality manner. Failure to ensure this throughout the implementation phase could result in changes in health outcomes for children and adolescents. The recommendations addressing these areas emerged from the individual contributions to the discussion made by participants in the focus groups and assessment workshops and have been summarized by the HIA project team.

## Figures and Tables

**Figure 1 ijerph-19-06428-f001:**
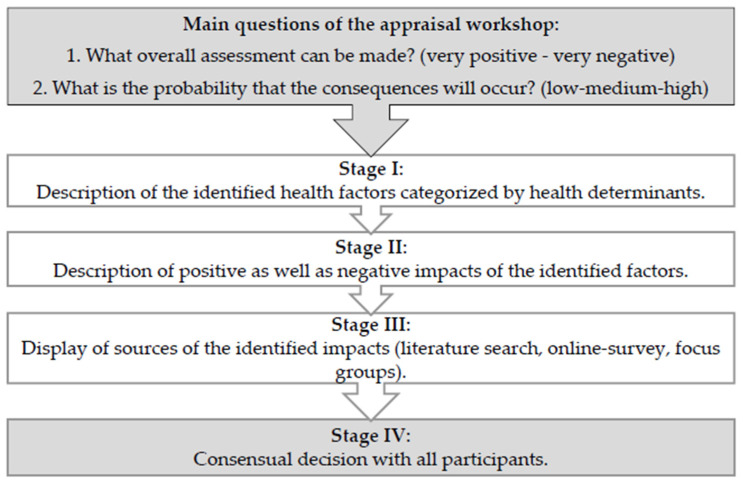
Graphical representation of the assessment process.

**Table 1 ijerph-19-06428-t001:** Methods for assessing potential health consequences.

Methods	Target Group	Content
**Literature review**	Children and adolescents (among others with attention on the particularly disadvantaged)	Research-based on the determinants of health
**Online-Survey**	School principals, teachers, administration, parents (representatives)	Assessment of the impact and concept
**Focus group 1**	Specialist inspectors for physical activity and sport	Impact and discussion of the scenario
**Focus group 2**	School principals	Impact and discussion of the scenario
**Focus group 3**	Teachers	Impact and discussion of the scenario
**Focus group 4**	Parents (representatives)	Impact and discussion of the scenario
**2× Appraisal workshop**	Experts	Assessment of the health impact and formulation of recommendations

**Table 2 ijerph-19-06428-t002:** Online focus group participants.

Focus Group Participants	Number of Participants
Specialist Inspectors for Exercise and Sport in Schools	*n* = 10
Principals/School Management/School Administration	*n* = 10
Parents/Legal guardians/Parent representatives (incl. statements collected from children/students)	*n* = 8
Teachers/Educators	*n* = 8

## Data Availability

Not applicable.

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
