# Peer review of "Health Impact Assessment (HIA) of a Daily Physical Activity Unit in Schools: Focus on Children and Adolescents in Austria Up to the 8th Grade"

_ijerph, 2022, doi:10.3390/ijerph19116428_

Round 1

Reviewer 1 Report

All of the comments are on the attached file. Please check the yellow and other color highlighted notes.

Author Response

Dear Sir or Madam,

Thank you for your valuable comments, all of which we have incorporated into the manuscript. I hope that the manuscript is now of high quality and I hope that the revisions meet your standards.

Please see the attachment regarding the point-by-point response.

Sincerely,

Madlene Movia

Reviewer 2 Report

Dear authors, thank for your paper, that is interesting and clear for readers - you should better explain the "4 health determinants" the you first mention at line 85 - how did you choose them? in which phase? so - complete the methodology with this specification. the second request - it is too complicate to go a little bit more in details about the assessment table  used for the appraisal workshop? - could you include it in the materials of the article as supplementary? Thank you

Author Response

(The authors gave the same response as above.)
